# Frozen Layers: Memory-efficient Many-fidelity Hyperparameter Optimization

**Timur Carstensen**[*,1,2] **Neeratyoy Mallik**[*,2,3] **Frank Hutter**[1,2,5] **Martin Rapp**[4]

[*]Equal contribution
[1]ELLIS Institute Tübingen
[2]Department of Computer Science, University of Freiburg, Germany
[3]Zuse School ELIZA
[4]Bosch Center for Artificial Intelligence (BCAI)
[5]Prior Labs

**Abstract** As model sizes increase, finding efficient and cost-effective hyperparameter optimization (HPO) methods becomes crucial for deep learning pipelines. Multi-fidelity HPO (MF-HPO) balances the computational resources needed for deep learning training with lower fidelity estimations. However, existing fidelity sources often struggle under lower compute and memory constraints. We propose a novel fidelity source: the number of layers trained or frozen during training. This approach offers significant savings in compute and memory for deep networks while maintaining rank correlations between hyperparameters at low fidelities compared to full model training. We demonstrate this in our empirical evaluation of ResNets and Transformers, analyzing the utility of frozen layers as a fidelity source in HPO and for a combined MF-HPO with other fidelity sources. This contribution opens new applications for principled MF-HPO using hardware resources as a fidelity and creates opportunities for improved algorithms that navigate joint fidelity spaces.

## 1 Introduction

In the last decade, we have observed considerable advances in deep learning (DL), which consequently has spread into nearly all domains of modern life. Several works have shown that the design of the training pipeline is the most important ingredient for obtaining high-performing DL models (Bello et al., 2021; Wightman et al., 2021). Typically, several hyperparameters (HPs) characterize the behavior of the training pipeline and are thus important to be tuned (Ruffinelli et al., 2020; Porian et al., 2024). Such HPs include optimizer parameters (e.g., learning rate, scheduler), regularization (e.g., weight decay), data augmentation, etc. However, optimal HPs depend on the specific setting (model, data, task) and need to be tuned separately for each (Snoek et al., 2012). The corresponding search spaces can be vast, as they grow combinatorially with the number of involved HPs. Therefore, efficient methods for *hyper-parameter optimization* (HPO) are needed.

The most resource-intensive step in HPO is the evaluation of an HP configuration, as this involves training a DL model, which can take hours or even days of GPU compute and require accelerators with a large VRAM (Cherti et al., 2023). *Multi-fidelity HPO* (MF-HPO) aims at reducing the resource requirements of such evaluations by leveraging cheap approximations of the training result (e.g., early validation loss or error). The core idea is that many HP configurations are evaluated at low fidelity (cheap), and only promising ones are further evaluated at higher fidelity (more resource-intensive) to find the optimal one (Jamieson and Talwalkar, 2016; Li et al., 2017; Falkner et al., 2018; Mallik et al., 2023). Thereby, the MF-HPO focuses resource usage on the most promising HP configurations. Existing multi-fidelity estimation techniques leverage the training duration (Li

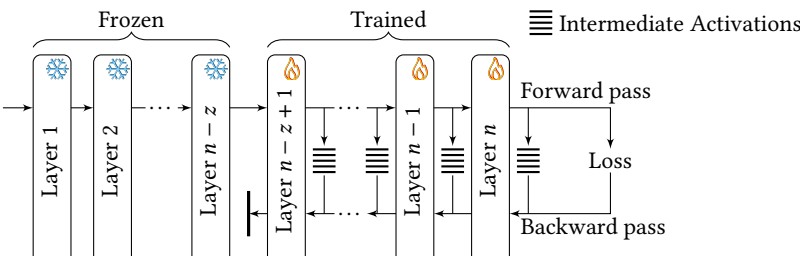

Figure 1: Training of a partially frozen neural network requires fewer resources: 1) lower compute due to a shorter backward path, 2) lower memory because the activations of the first $z$ layers are not kept in memory for the backward pass, and 3) lower memory due to no optimizer states for the first $z$ layers. The resources requirements are adjustable via $z$.

et al., 2018), dataset size (Klein et al., 2017), or model size (Falkner et al., 2018; Swersky et al., 2014). However, they cannot effectively reduce all involved resources, i.e., GPU compute *and* memory. Kandasamy et al. (2017, 2020) presented a *Multi-fidelity Bayesian Optimization* algorithm that can handle multiple sources of fidelity, however, their experiments only focus on compute resources.

In this work, we present a novel fidelity that is based on varying the number of trainable layers in a DL model and freezing others, as illustrated in Figure 1. This allows us to *partially* evaluate a model, moving away from the typical black-box nature of model training in HPO. Freezing significantly reduces the resource requirements for training, both in terms of GPU compute and GPU memory, reducing each by $\geq 2\times$, depending on the model configuration and batch size (see Fig. 2).

These savings come at the cost of noisier estimates of the performance of HP configurations since frozen layers stay at random initialization throughout training. However, it has been shown that randomly-initialized DL models can extract powerful features even without training and exhibit similar inductive bias as the fully trained network (Saxe et al., 2011; Gaier and Ha, 2019; Zhong and Andreas, 2024). We observe that, given a fixed set of frozen layers, and thus random features to finetune the later layers to, the relative performance of different HPs correlate strongly. For example, training with only the final half of the layers already yields a reliable indicator of relative hyperparameter performance, when compared to training the full model. We also show that this threshold can be improved for architectures with strong inductive biases. These reliable approximations can strongly reduce the required GPU compute and memory for estimating the performance of HP configurations. The frozen layer count can either server as an alternative fidelity source in multi-fidelity hyperparameter optimization (HPO) or be used in conjunction with existing fidelity measures.

In summary, we make the following novel contributions in this work:

- We introduce *layer freezing* as a novel source of fidelity for HPO of deep learning models. We demonstrate both its eligibility as an effective fidelity and its unique capability to offer memory savings for low fidelity evaluations. This is the first multi-fidelity source that *opens* the DL model training process itself, i.e., it modifies gradient computation and weight update steps.

- We demonstrate that common deep learning architectures, with some or most layers frozen, exhibit strong HP rank correlations compared to the full model training, making layer freezing a strong source of approximations in MF-HPO.

- We empirically show that layer freezing as fidelity along with the training budget, offers opportunities for novel search algorithms for improved cost and memory savings in MF-HPO.

## 2 Related Work

The broad scope of this paper is HPO, which falls under the AutoML umbrella (Feurer and Hutter, 2019). However, given the scope of our contribution, we specifically consider HPO for large-scale DL, which requires careful design for compute resource management.

**Hyperparameter Optimization**. HPO for DL can be seen as a bilevel optimization where the inner loop is the actual task to be solved using stochastic gradient descent, and the outer loop is designed to optimize the variables that impact the inner task and how the task is solved. This is an iterative optimization loop with usually a global budget for the entire optimization process. When this inner task is a complete *black-box* and can only receive inputs (its hyperparameter configuration and training budget) and return a corresponding performance metric, the outer-HPO loop can be solved using *black-box optimization.*

Bayesian Optimization (BO) (Snoek et al., 2015; Frazier, 2018; Cowen-Rivers et al., 2022; Garnett, 2023) and Evolutionary algorithms (EA) (Real et al., 2019; Awad et al., 2020) are strong global optimizers that handle black-box optimization reasonably well. However, the nature of the task can yield different behaviors and BO and EA methods could require many evaluations to reach good performance.

Given the current growth in model sizes commonly used in practice (Kaplan et al., 2020; Hoffmann et al., 2022), *online*-HPO is still expensive, especially when treating the DL task as a black-box. This assumption can be relaxed when the HPO algorithm can query or control the *inner* DL task for cheaper, approximated evaluations. Most commonly one trains a DL model on only a subset of the training data to approximate performance. Since the DL task can now be intervened in and queried before full training convergence, this is known as the *gray-box* view of the inner task in the bilevel HPO loop (Feurer and Hutter, 2019; Astudillo and Frazier, 2021). This is more commonly known as *multi-fidelity* HPO and can be seen as a direct extension of the usual black-box methods with different fidelity sources. Systems have been developed to manage these multi-fidelity trials by dynamically allocating computational resources in cloud settings to maximize model accuracy under strict time and monetary budgets (Dunlap et al., 2021). In this work, we introduce additional fidelity dimensions that enable efficient approximation of full training costs and model evaluation metrics.

Even though a single fidelity source can offer significant compute trade-offs during HPO, combining multiple fidelity sources does not necessarily compound the cost savings and is likely to have diminishing returns. Further, not many works in the literature have reported studies using multiple sources of fidelity together, especially for modern large-scale DL (Kandasamy et al., 2017, 2020; Siems et al., 2020; Bansal et al., 2022). However, *gray-box* approaches still intervene at the level of the training routine and not the model directly. One must look inside the black- or gray-box to make it a *glass-box* and thus take better HPO decisions. Dynamic Algorithm Control (Adriaensen and Nowé, 2016; Adriaensen et al., 2022) is a field which aims to look inside the optimization process, but is yet to effectively handle large-scale DL applications. Similarly, gradient-based HPO treats the entire bilevel HPO problem as an end-to-end task to be optimized (Franceschi et al., 2017; MacKay et al., 2019; Lorraine et al., 2020). While these methods are efficient in terms of the number of unique model trainings required, they face a critical limitation: they demand significantly more memory and compute, with requirements that grow proportionally with model size.

Our contribution directly addresses this scalability gap by introducing a novel fidelity for cheaper tuning of DL models. Specifically, we show that by strategically determining which layers are frozen or trained during optimization, we open up the black-box in a computationally efficient manner. This approach provides strong, reliable signals to the outer-level HPO task while substantially reducing computational costs.

**Model-growing**. Architectural approximations are not a new approach for efficient tuning and training of DL models (Brock et al., 2017; Dong et al., 2018; Liu et al., 2018; Hu et al., 2019). Weight-

sharing based super-net training can also be seen as effectively training a smaller subset of the parameters (Liu et al., 2019; Cha et al., 2022). Similarly, there may be parallels with why certain models can be pruned post-training while retaining the learned function. However, this sub-field of Neural Architecture Search (NAS) is specific to finding new architectural designs, with or without additional constraints, usually given a set of training HPs. Our work focuses on the task of tuning HPs (and potentially architectural components) given a predetermined target model architecture. Thus, the scope of our work is different from these prior works. In principle, individual models, at any scale, can even be tuned using our fidelity, and can then be grown as desired (Gong et al., 2019; Shen et al., 2022; Wang et al., 2023; Du et al., 2024; Samragh et al., 2024; Yao et al., 2024; Mallik et al., 2024). Our work is thus orthogonal and can jointly be applied with these techniques.

## 3 Layers as Fidelity

In this section, we describe the problem scope and formulation and introduce our main contribution.

### 3.1 The HPO Problem

The HPO algorithm aims at minimizing an expensive function $f(\boldsymbol{\lambda})$, i.e., finding $\boldsymbol{\lambda}^* = \arg\min_\lambda f(\boldsymbol{\lambda})$. Here, $\lambda \in \Lambda$ is a HP configuration from a search space of HPs $\Lambda$, that control and determine the behavior of $f(\cdot)$, the DL task at hand. This is especially the *black-box* view, where the search for $\lambda^*$ is agnostic to the inner-task $f(\cdot)$. More concretely, if $f(\cdot)$ represents training a deep neural network, instantiated with HPs $\lambda$ (learning rate, weight decay, etc.), then $f(\lambda)$ represents the evaluation measure, for instance, the validation loss given a fixed training budget. The HPO searches for the $\lambda$ with the lowest validation loss, given a larger, global HPO budget.

When this inner task allows intermediate queries, early-stopping, and resuming of the evaluations, the optimization of the inner task is treated as a *gray-box*. This can be denoted as, $f(\cdot) \approx \hat{f}(\cdot, z)$, where $z$ represents the fidelity variable, and $\hat{f}(\cdot)$ represents the approximation of the true task given a fidelity. Such HPO formulations are called *Multi-fidelity* HPO, as $\hat{f}(\lambda, z < z_{\max})$ approximates the target evaluation of $\hat{f}(\lambda, z = z_{\max}) \rightarrow f(\lambda)$, for a much lower cost for the evaluation: $cost(\hat{f}(\lambda, z < z_{\max})) \ll cost(\hat{f}(\lambda, z = z_{\max}))$. Here the $cost(\cdot)$ could be any function returning a compute estimate which has a monotonic relation with $z$. Note, typical fidelities are bounded, $z \in [z_{\min}, z_{\max}]$ and are a feature of the task or application at hand. With our layer freezing, $z_{\max} \in \mathcal{N}$ corresponds to the full number of layers. Therefore, $\hat{f}(\lambda, z = z_{\max}) \rightarrow f(\lambda)$ represents training all layers, instantiated and trained based on $\lambda$. In this work, we design fidelity approximations $z$ for DL tasks that satisfy the assumptions underlying MF-HPO.

To the best of our knowledge, we are the first to *open* the DL model or the task $f(\cdot)$ to access an explicit source of fidelity, $z_{\text{layers}}$, for MF-HPO. Next, we define key properties that constitute valid fidelity approximations for MF-HPO in DL contexts.

**Fidelity Formalism**. For a variable to serve as a valid fidelity parameter in DL tasks, it must satisfy the following essential properties:

1. *Cost monotonicity*: for $\lambda$ and $z_1 < z_2 \leq z_{\max}$, then $cost(\hat{f}(\lambda, z_1)) < cost(\hat{f}(\lambda, z_2))$

2. *Mutual Information monotonicity*: evaluation on a higher fidelity should inform more, as $\mathcal{I}(f(\cdot); \hat{f}(\cdot, z_1)) < \mathcal{I}(f(\cdot); \hat{f}(\cdot, z_2))$ for $\forall z_{i,j} \in [z_{\min}, z_{\max}]$, and $i < j$ [1]. In practical applications, rank correlation is used as a proxy metric: for two HPs $\lambda_1, \lambda_2$ where $f(\lambda_1) < f(\lambda_2)$, and fidelity $z < z_{\max}$, rank correlation holds if $\hat{f}(\lambda_1, z) < \hat{f}(\lambda_2, z)$ with high probability $p > 1 - \delta_z$, with $\delta_z \ll 1$. Higher fidelity increases rank correlation: $\delta_{z_1} > \delta_{z_2}$ for $z_1 < z_2$.

We now detail how our layer-based fidelity approach satisfies these formal requirements.

---

[1] Relaxed assumption, since in practice, $\mathcal{I}(f(\cdot); \hat{f}(\cdot, z))$ may not be symmetric for multi-fidelity evaluations.

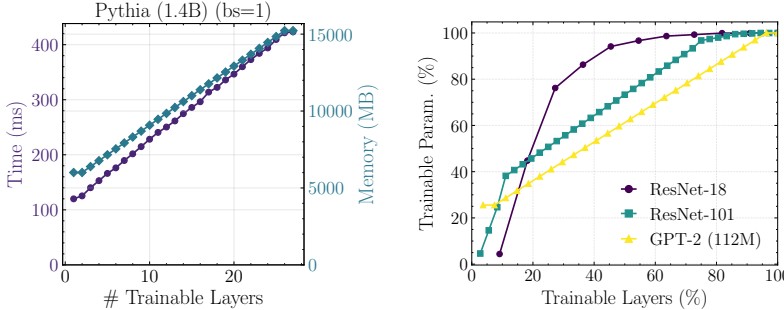

Figure 2: (*Left*) Hardware resource requirements for Pythia 1.4B (Biderman et al., 2023) with batch size 1 at each fidelity. Time refers to the time taken per step (forward, backward, and optimizer step). At the lowest fidelity, our method reduces memory requirements by a factor of $\geq 3\times$ and speeds-up runtimes by a factor of $\geq 4\times$. (*Right*) Comparison of trainable parameters under layer freezing as fidelity. Different architectures distribute parameters unevenly across layers, resulting in varying computational costs as fidelity increases.

### 3.2 Freezing layers for Multi-fidelity HPO

The main idea of this work is to use the number of frozen or trainable layers in a network to adjust the fidelity. Given a neural network $M$ with $n$ layers and the desired number of trained layers $z$, we want to freeze the first $n - z$ layers and train the remaining $z$ ones, as shown in Figure 1. Below, we explain why layers as fidelity satisfies the monotonicity requirements and the likelihood of strong rank correlation of HPs.

**Impact on resource requirements**. Partial freezing comes with major resource savings:

- No backward pass (weight gradients) needs to be computed for the frozen layers, reducing the required computational resources.

- Intermediate activations in the frozen layers do not need to be kept in memory for the backward pass, reducing the required memory resources.

- No optimizer state needs to be kept in memory for frozen weights, further reducing the memory requirements. This becomes especially effective for optimizers such as Adam(W) (Kingma and Ba, 2015; Loshchilov and Hutter, 2019) which track two running estimates per model parameter.

These properties makes the number of frozen layers as a fidelity unique in its resource efficiency. Given that such a setup directly tunes HPs for the full target model, training lower fidelities of a large model on smaller hardware is now possible (see Fig. 6a). The resource savings or compute cost is strictly monotonic under trainable layers as fidelity (see Fig. 3).

**Impact on performance**. The first $n - z$ layers remain at their random initialization, providing random features for the later layers to learn from. We expect that, given a particular number of frozen layers, the network's capacity to express a function is bounded. With more trainable layers, we expect this capacity to increase. Therefore, given a fixed HP, trained for a similar budget, training more layers will *likely* not produce a worse loss than training fewer layers. Thereby, performance across fidelity is likely to be monotonic. However, in MF-HPO we are interested in the low fidelity evaluations being *informative* signals or proxy for the full evaluation (Section 3.2). Depending on the task, a certain proportion of frozen layers, random features, and trainable last layers, is adequate to express the network capacity, especially for a relative ranking of HP performance. This is what allows layer-freezing to be used for multi-fidelity HPO.

**Continuation changes task**. Unlike other black- or gray-box fidelity sources, such as epochs or data subsets that can easily restore or continue training a saved checkpoint for higher fidelity (more

epochs or data), the same cannot be achieved for the number of trainable layers without using heuristics that may not generalize well. Layer-freezing with *continuations* would be equivalent to training a network with a schedule for unfreezing or freezing layers. Naturally, the meaning of *good* HPs for the DL task may change given that such schedules will likely interact with other HPs influencing training such as learning rate schedule. Therefore, we consciously do not pursue this direction of *continuing* over layers as fidelities and dedicate it to its own focused future work.

### 3.3 How to choose Layers

There are a few practical considerations to make when discretizing an architecture into separate layers or blocks. We generally follow the following rules-of-thumb which can be applied to most architectures without loss of generality, to split architectures into *layers* suitable for use as fidelity parameters:

(i) First, decompose the architecture into all sequential sub-components, ignoring skip connections. This establishes the maximum granularity of layers possible as a fidelity.

(ii) If literature exists for the architecture, leverage established groupings of these sequential sub-components to define meaningful layer indices (see Fig. 13 for Transformer example).

(iii) In the absence of prior information, establish layer group boundaries at non-linear activations, ensuring each layer either begins after or ends with such an activation.

(iv) For excessively deep networks, consolidate identified layers into larger functional blocks (e.g., ResNet residual blocks). Ensure that each discrete fidelity increment corresponds to a substantial change in parameter count and training cost.

We note that every architecture design will have its nuance in how to split and there may be an optimal splitting per architecture. However, in this work, we only aim to explore *if* layers as fidelity can be practical, and the optimal algorithm for layer splitting is left for future work. We refer to Appendix F for more details on how we split the architectures considered in this paper. In the next section, we apply the above rule-set to split the architectures in our experimental setup, for a general approach. We demonstrate through empirical validation that it *is* feasible to split architectures into layers that serve as effective fidelity approximations for HPO.

## 4 Experimental Evaluation

In this section, we experimentally validate our proposed layer-freezing approach across two architectural families. Through comprehensive HP sweeps, we evaluate whether layer freezing satisfies the formal requirements of a fidelity measure (see Section 4.2) and quantify its practical benefits for multi-fidelity HPO. Our experiments address three key questions: (1) Does layer freezing provide significant cost savings? (2) Does it maintain strong rank correlations with full-model performance? and (3) How effectively can it be combined with other fidelity sources?

### 4.1 Setup

We use models from two architectural families: GPT-2-style Transformers (Radford et al., 2019; Vaswani et al., 2017) and ResNet (He et al., 2016), using the implementations from LitGPT (AI, 2023) and torchvision (TorchVision maintainers and contributors, 2016), respectively.
Experimental details for each architecture type can be found in Appendix B. The hyperparameter search space for our parameter sweeps is reported in Table 1. We evaluate performance using Spearman's rank correlation across the entire hyperparameter grid. In our experiments, full-fidelity performance is obtained by training models with all layers trainable until convergence. For hardware metrics, we measure both GPU memory and runtime under different configurations of trainable layers and batch sizes (see Appendix E for detailed measurement methodology).

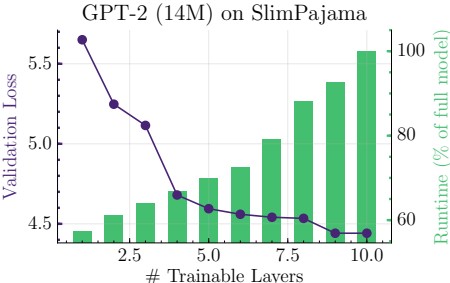 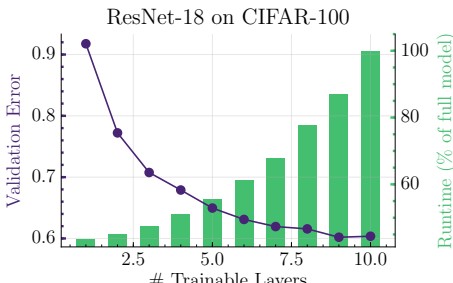

Figure 3: The *x*-axis shows the discrete number of layers being trained, starting from the output moving backwards. The highest number of trainable layers represent full model training, therefore, best performance and most cost incurred. Runtime is how long a single step (forward + backward + optimizer step) takes at each fidelity when compared to the fully trainable model. *(Left)* 14M parameter GPT-2 model trained for 20 tokens per parameter at each fidelity. *(Right)* ResNet-18 trained on CIFAR-100 for 20 epochs at each fidelity.

## 4.2 Layers as a fidelity

Here, we evaluate frozen layers as fidelity following the formalism established in Section 3.2. For Figure 3, we randomly select a configuration from our sweep and observe the validation loss obtained under the same training budget but with varying numbers of frozen layers after random initialization. We note that each architecture exhibits a distinct distribution of cost savings and performance variations, representing different approximation noise characteristics. However, crucially, both performance, compute cost, and rank-correlation (see Fig. 4) follow a monotonic trend across all architectures, satisfying a fundamental requirement for frozen layers to serve as a valid approximation for full model training. This monotonic relationship holds consistently across architectures. In Appendix E we show other cost metrics to show the monotonic cost gain with frozen layer fidelity. We also perform an ablation on the direction of (un-) freezing (Appendix F.2). The results show that one must unfreeze starting from the output layers to benefit from practically any compute savings at lower fidelities.

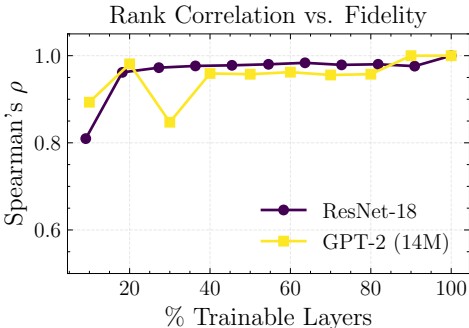

Figure 4: Rank correlation with full-fidelity validation performance for a 14M parameter GPT-2 and ResNet-18. Each hyperparameter configuration received the same training budget. Refer to Table 1 for the search space of configurations. Each evaluation is treated as a *black-box* evaluation given trainable layers as fidelity.

## 4.3 Layers as fidelity in MF-HPO

Similarly, to the previous section, each run received the same training budget in terms of the number of update steps given a hyperparameter, across all discrete fidelity levels. At each fidelity,

we compare the final performance for all hyperparameter configurations in the respective grids for each architecture. Figure 4 shows the rank correlation at each fidelity, compared to the rankings for the full model training (100%). Remarkably, for the search spaces in discussion, training up to 40% of the layers is adequate for a rank correlation $\approx 1$. Comparing this with numbers from Figure 3 shows, that the degree of gains will be varied but significant, if a hyperparameter can be reliably zero-shot transferred by training less than half the layers.

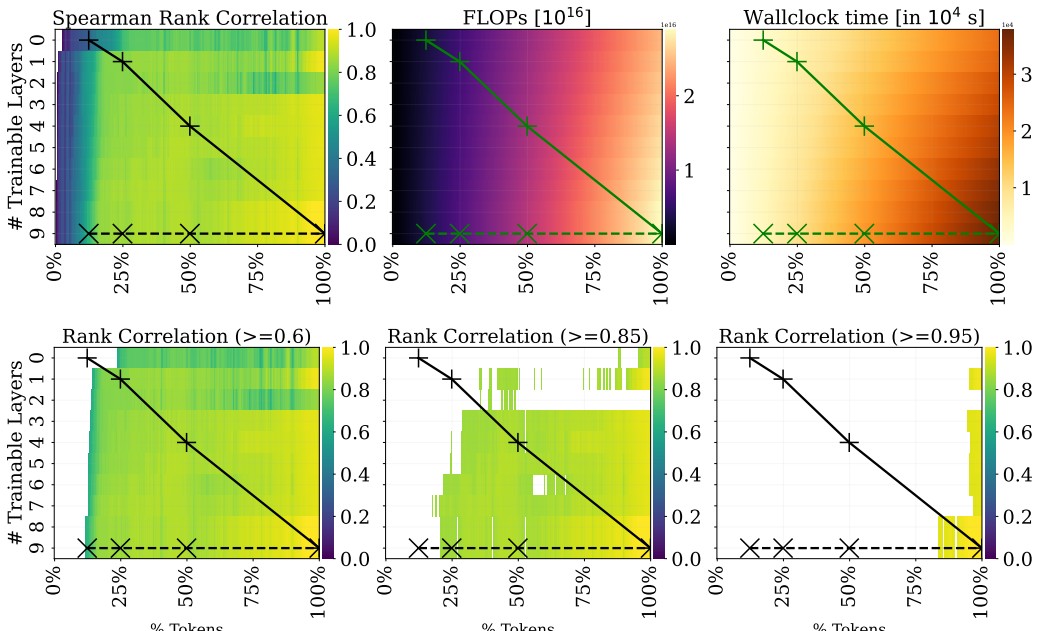

Figure 5: Hyperparameter rank correlation landscape across the joint fidelity space of trainable layers (y-axis) and training tokens (x-axis) for a 14M GPT-2 model (hyperparameter details in Table 1). Two black traces represent potential *Successive Halving* (SH) runs with $\eta = 2$: the dashed line shows traditional SH using only data as fidelity, while the solid line demonstrates our proposed approach using both layers and data as fidelities. Markers indicate SH query points, with joint fidelity queries at (1 layer, 12% tokens), (2 layers, 25% tokens), (5 layers, 50% tokens), and single fidelity queries at (all layers, {12, 25, 50}% tokens). The three *bottom-row* plots visualize rank correlation thresholds of {0.6, 0.85, 0.95} respectively. Notably, except at the lowest fidelity where additional layers provide stronger correlations, a joint fidelity approach achieves better correlation with reduced computational cost in both wall-clock time and FLOPs (see Table 3 for quantitative comparisons).

**Many-fidelity HPO**. Layer-freezing offers itself as an orthogonal fidelity type and can thus be used in conjunction with other usual fidelity sources. For sub-epoch language model training, where data doesn't repeat, the notion of epochs and data sub-samples blend into one fidelity. We use our frozen layers as a fidelity, along with training budget in the form of tokens seen in Figure 5. We plot the hyperparameter rank correlation landscape across the joint fidelity space, with discrete layers and uniformly discretized training steps, and the cost landscape for wall-clock time and FLOPs consumed. The *top* row in Figure 5 shows the rank correlation and cost landscapes in the joint fidelity space for the given parameter sweep. The solid and dashed lines represent two traces for *Successive Halving* with two and single fidelity variables respectively, under $\eta = 2$ (Jamieson and Talwalkar, 2016). The *bottom* row in Figure 5 shows the rank correlation landscape but only those above a given threshold.

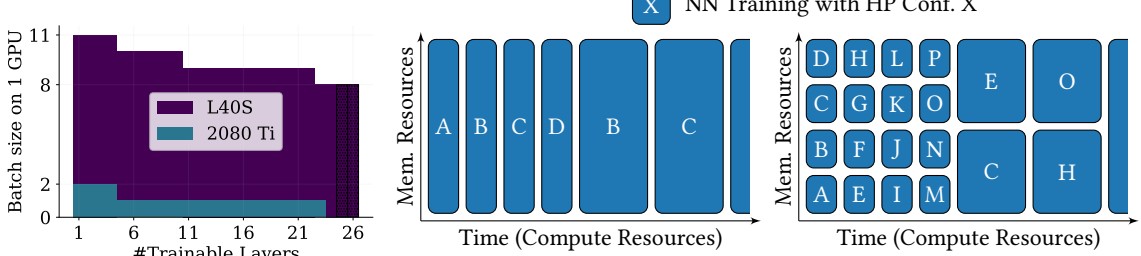

(a) Layer freezing enables training DL models on weak hardware.

(b) Left: Classical successive halving with existing low-fidelity estimation. Right: Memory-parallel successive halving with partial freezing (ours)

Figure 6: *(a)* Batch size comparison for a 600M parameter GPT2 model (24 hidden layers) The embedding and un-embedding layers are tied (shared weights) and hence the full fidelity and the (full fidelity - 1) are effectively the same runs (marked with dots). **Freezing layers naturally saves memory, which can allow successful HPO for a large model on much cheaper hardware**. *(b)* Freezing-based multi-fidelity estimation can be integrated into successive halving by leveraging memory-parallel training.

The key observation is that for a low number of trainable layers, given adequate data, we can have extremely high-rank correlations (Fig. 5, bottom, right). Especially when looking at columns 2 and 3 together, we see that there is potential to be in high rank correlation regions for much lower compute and wall-clock time, by training much fewer layers. This also has the practical benefit of requiring far less memory to tune low fidelity models. Alternatively, run more low fidelity evaluations utilizing the memory savings (see next, Section 4.4).

These findings have significant implications for HPO of large-scale models. Our layer-freezing approach enables tuning on memory-constrained hardware that would otherwise be infeasible while maintaining strong rank correlations even at low fidelities. This suggests that specialized HPO algorithms designed to navigate this joint fidelity space could achieve substantial efficiency gains, opening new possibilities for cost-effective tuning of increasingly complex architectures.

## 4.4 Leveraging Reduced Memory Consumption in MF-HPO

Our experiments focus on demonstrating the efficacy of using frozen layers as fidelity in *Multi-fidelity* HPO. The empirical gains of using this fidelity source is in its memory savings and that can be realized in two ways. In Figure 6a we show an example where using frozen layers as fidelity can effectively allow using GPU resources as fidelity. This has important practical implications as computing clusters with a mix of hardware resources tend to follow a pyramidal structure of more nodes of low, cheaper resources and fewer of the high memory and bandwidth GPUs. *Successive Halving*-like algorithms follow a similar design to trade off the number of configurations and the fidelity at which to evaluate them.

Alternatively, given high-end hardware, the low memory requirement under training with frozen layers implies that more memory is available to run more of such low fidelity evaluations. Modern GPUs allow for sharding and slicing[2], which can in principle be used to simulate more GPU resources (see Fig. 6b). Frozen layers as fidelity offer a ready solution to adapt *Multi-fidelity* HPO to such hardware-specific DL tuning.

## 5 Conclusion

We introduced layer freezing as a novel fidelity source for HPO, formalizing the requirements for valid fidelities in deep learning contexts—the first such rigorous analysis for MF-HPO in deep

---

[2]Recent versions of NVIDIA GPUs allow to use a variable number of independent GPU slices with MIG mode. This effectively allows evaluation of multiple HP configurations at a low fidelity in parallel without resource contention.

learning. Our approach satisfies these formal criteria while offering unique memory savings and maintaining strong rank correlations with full-model performance. Experimental results across two architecture families demonstrate that even with a significant portion of layers frozen, the relative performance of HP configurations is preserved, enabling the tuning of large models on memory-constrained hardware. This opens new directions for memory-efficient multi-fidelity HPO algorithms that can effectively navigate joint fidelity spaces.

**Limitations**. Despite the promising results on commonplace architectural choices, there could be practical limits. The optimal layer discretization strategy is architecture-dependent and requires domain knowledge or heuristics to implement effectively, especially for a new or specialized architecture. The effect of layer-specific HPs on frozen layers as fidelity should be studied along with a wider benchmarking. Our approach currently lacks a principled continuation mechanism that is crucial to enable freeze-thaw MF-HPO. This leaves room for compute savings since each fidelity evaluation with frozen layers requires a new training from scratch. However, new specialized architectures could have strong inductive biases that further improve low fidelity rank correlations with frozen layers, ameliorating the need for continuation.

## 6 Broader Impact Statement

Our layer-freezing approach democratizes access to deep learning by enabling model tuning on more affordable hardware, lowering barriers to entry for researchers with limited computing resources. The improved compute- and memory-efficiency also contribute to reduced environmental impact through lower energy consumption.

**Acknowledgements**. This research was partially supported by the following sources: EU Project ELSA under grant agreement No. 101070617. TAILOR, a project funded by EU Horizon 2020 research and innovation programme under GA No 952215; the Deutsche Forschungsgemeinschaft (DFG, German Research Foundation) under grant number 417962828; the European Research Council (ERC) Consolidator Grant 'Deep Learning 2.0' (grant no. 10). This research was partially funded by the Deutsche Forschungsgemeinschaft (DFG, German Research Foundation) under grant number 539134284, through EFRE (FEIH 2698644) and the state of Baden-Württemberg. Neeratyoy Mallik is supported by the Konrad Zuse School of Excellence in Learning and Intelligent Systems (ELIZA) through the DAAD programme Konrad Zuse Schools of Excellence in Artificial Intelligence, sponsored by the Federal Ministry of Education and Research. Frank Hutter acknowledges financial support by the Hector Foundation. The authors acknowledge support from ELLIS and ELIZA. Funded by the European Union. Views and opinions expressed are however those of the author(s) only and do not necessarily reflect those of the European Union or the ERC. Neither the European Union nor the ERC can be held responsible for them.

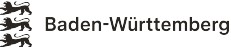 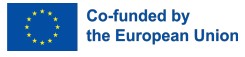

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

## A Search Spaces

Additional information about search spaces are shown in Table 1.

Table 1: Considered hyperparameter search spaces for ResNet and Transformer in Section 4.

| Hyperparameter | Range/Options | Architecture |
|---|---|---|
| *Optimization* | | |
| Learning Rate | $\{10^{-4}, 10^{-3}, 10^{-2}\}$ | ResNet |
| Weight Decay | $\{10^{-5}, 10^{-4}, 10^{-3}, 10^{-2}\}$ | ResNet |
| Weight Decay | $\{0, 10^{-6}, 10^{-4}, 10^{-2}\}$ | Transformer (14M) |
| Optimizer | {Adam, SGD} | ResNet |
| $\beta_1$ | {0.9, 0.95, 0.99} | ResNet |
| $\beta_2$ | {0.9, 0.95, 0.99} | ResNet |
| $\beta_1$ | {0.9, 0.95} | Transformer (14M) |
| $\beta_2$ | {0.95, 0.99} | Transformer (14M) |
| *Learning Rate Schedule* | | |
| Warmup Fraction | {0.05, 0.1, 0.25} | Transformer (14M) |
| Cooldown Fraction | {0.1, 0.25, 0.5} | Transformer (14M) |

## B Model details

Since the ResNet implementation that we used was the one provided in `torchvision` (TorchVision maintainers and contributors, 2016), we only provide details on our language model implementation here. We build on the implementation by `LitGPT` (AI, 2023) and only extend the model definition by adding more logging utilities. The architectural parameters for the models used are shown in Table 2.

Table 2: Architectural parameters of the language models that were used in our experiments.

| Parameter | 14M Model | 600M Model | Pythia 1.4B |
|---|---|---|---|
| Embedding Dimension | 128 | 1280 | 2048 |
| Number of Layers | 8 | 24 | 24 |
| Number of Heads | 2 | 20 | 16 |
| Context Length | 1024 | 1024 | 2048 |
| Vocabulary Size | 50257 | 50257 | 50257 |
| Normalization | LayerNorm | LayerNorm | LayerNorm |

Apart from the parameters mentioned in Table 2, we use the default parameters in `LitGPT`.

## C Additional results on rank correlation

## D Additional results on performance monotonicity

Additional results on performance monotonicity are shown in Fig. 8.

## E Hardware Metrics

**Data collection setup**. We collect both runtime and memory metrics for various combinations of batch sizes and architectures. For all considered combinations, we collect metrics for all fidelities (i.e., from one trainable layer to the fully trainable architecture). Measurements are taken on compute nodes with NVIDIA L40S GPUs with 48GB of VRAM. To account for variability in measurements we perform a number of warm-up passes before recording metrics. We typically perform 100 warm-up and another 100 measurement passes at the specified batch size and report the mean. Results are shown in Fig. 9.

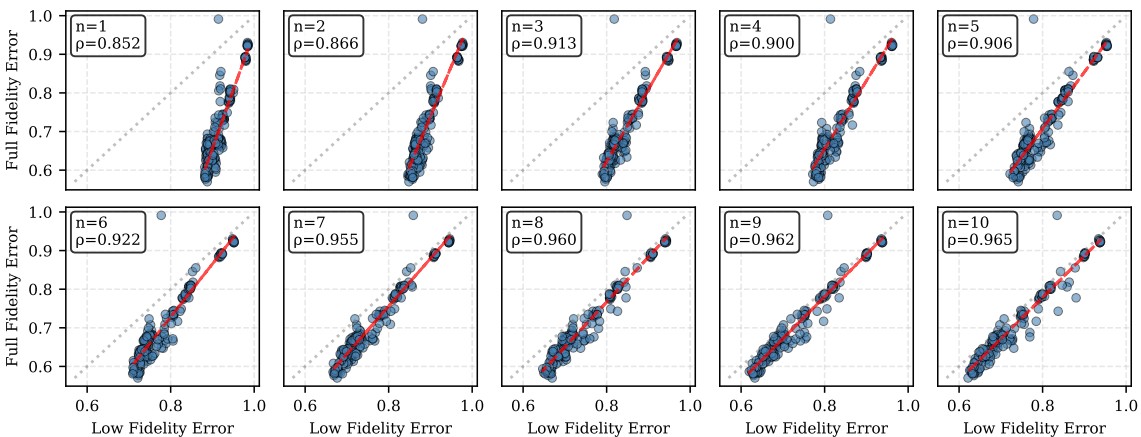

Figure 7: Low-fidelity validation errors for ResNet-18 trained on CIFAR-100 for 20 epochs vs. the full fidelity validation error. Even at the lowest fidelity, rank correlation is quite high. $n$ is the number of trainable layers with $n = 11$ being the full fidelity.

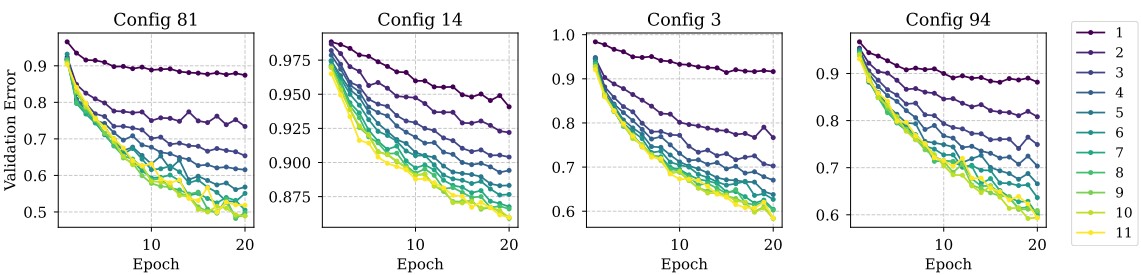

Figure 8: Validation error trajectories for ResNet-18 on CIFAR-100 across trainable layers and epochs. Each subplot depicts the validation loss trajectory of a randomly sampled HP configuration. Performance is highly monotonic in the number of trainable layers over the epochs considered.

## F  Layer-freezing Technique

Our layer-freezing technique is described in Algorithm 1. To make our algorithm work with as many neural architectures as possible, we make few assumptions about the structure of the model's forward pass. In particular, we assume that the top-level forward-pass of the architecture is sequential, i.e., that inputs are processed sequentially by each layer. However, each layer may itself contain various sub-layers which may have a different forward-pass structure (e.g., skip connections). Hence, by default, we only split layers in the architecture that are contained in a `nn.Sequential` container. We leave it up to the user to pass further sub-layer classes to our algorithm that are supposed to be split via the $U$ argument of Algorithm 1. This allows the user to define the granularity of the layer-splitting and freezing.

Given the simplicity of our algorithm, the implementation is very high-level and has no external dependencies apart from PyTorch. The user-facing API is very minimal and allows the use of our fidelity with a simple one-line code change.

### F.1  Details on architecture splitting

For the two considered architectures, Transformers and ResNets, we draw parallels to how the models are defined in PyTorch code to discretize. That is, the larger functional blocks in each architecture's definition (in code) also provide us with our discretization granularity.

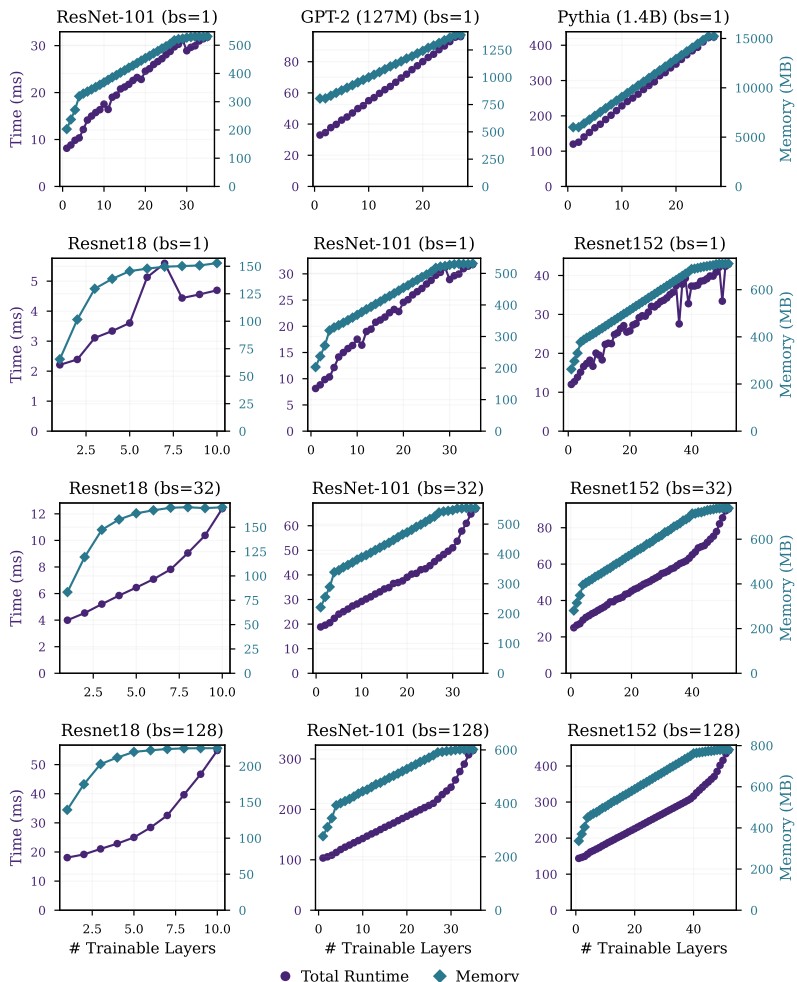

Figure 9: Hardware-performance metrics across different numbers of trainable layers (shown replicated vertically). Memory is the amount VRAM allocated to the respective Python process on the GPU (see Appendix E for more details). Total runtime refers to the time taken to complete one iteration (forward pass + backward pass + optimizer step).

**Transformer**. We discretize the architecture into input embeddings, Transformer Encoder blocks, and output embeddings. In small models (shallow and/or narrow) with large vocabularies, the input- and output embeddings constitute a large part of total parameters. In particular, our 14M parameter GPT-2 model, used throughout most of our experiments, has about 6M parameters solely in the embeddings. Hence, the lowest fidelity incurs a relatively high cost to train with the intermediate fidelities providing a more gradual increase in cost (see Fig. 2). The oversized effect of the embeddings decreases as one increases the model width and depth.

**ResNet**. We discretize all ResNet-style architectures by the output projection as well as inverted bottleneck blocks. As ResNets have a large amount of their parameters in the last few layers, the first few fidelities lead to quite high jumps in the cost associated with training at these fidelities but this evens out relatively soon, depending on the depth of the architecture (see Fig. 9).

**Algorithm 1** Layer-splitting and freezing technique

---

**Require**: model $M$, number of trainable layers $n_{trainable}$, optional unwrap class types $U$
1:  $all\_layers \leftarrow []$                                         ▷ Initialize empty list to store all layers
2:  **function** RecursiveTraversal($module$, $U$)
3:      **if** $module$ is instance of $U$ or Sequential **then**
4:           **for** each $child$ in $module.children()$ **do**
5:                RecursiveTraversal($child$, $U$)
6:           **end for**
7:      **else**
8:           append $module$ to $all\_layers$
9:      **end if**
10: **end function**
11: RecursiveTraversal($M$, $U$)
12: $param\_layers \leftarrow$ filter $all\_layers$ to only include those with parameters
13: $trainable \leftarrow$ last $n_{trainable}$ layers from $param\_layers$
14: $frozen \leftarrow$ all layers from $param\_layers$ except those in $trainable$
15: FreezeParameters($frozen$)
16: UnfreezeParameters($trainable$)
17: **return** $frozen$, $trainable$

---

### F.2 Freezing order

We conducted an ablation to test whether our intuition that the freezing order matters greatly is correct. In Figure 10 we show the difference in the per-iteration runtime for training a 127M GPT2 Transformer model when the unfreezing direction is either from the output (our method) vs. the input layers of the network.

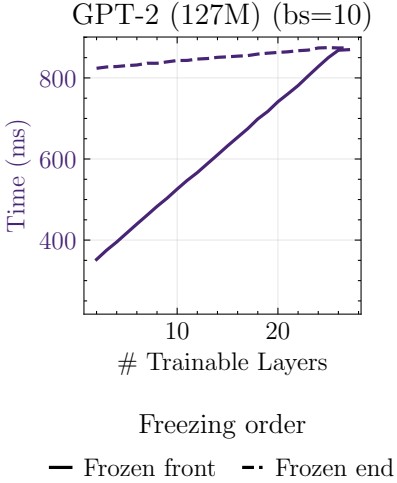

Figure 10: Runtime (one training step) for a 127M GPT2 Transformer when freezing starting from the input layers vs. starting from the output layers.

The results corroborate our hypothesis. When making the layers closest to the input trainable, the backward pass has to propagate all the way to the front of the network, effectively eliminating most of the compute savings. We see a slight decrease in runtime even when unfreezing from the input layers which is attributable to the reduced number of parameters and parameter groups that need to be updated by and in the optimizer. The memory requirements (not show) are also larger

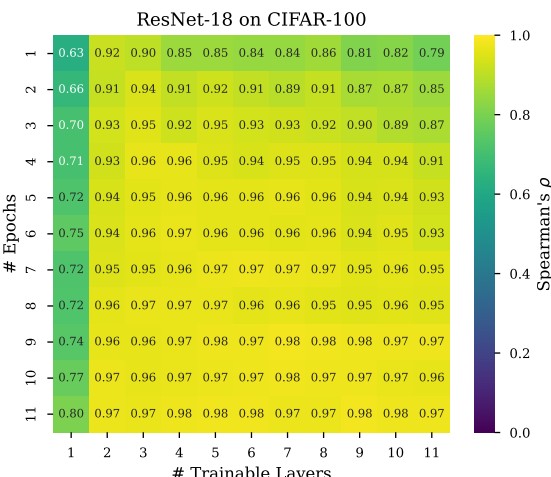

Figure 11: Rank correlation with the full fidelity (11 trainable layers, 20 epochs on CIFAR-10 [not shown here]). Each cell shows the rank correlation for a particular setting of epochs and trainable layers.

when unfreezing from the front since the activation maps of later, frozen layers, need to be kept to be able to compute the gradients of the unfrozen layers closer to the input.

## G Multi-multi Fidelity

To perform our analysis of a multi-multi-fidelity HPO setting, we study the combination of layers and epochs as fidelities. Table 3 shows the wall-clock time and FLOPs speed-ups / savings for a diagonal schedule for both fidelities (increasing both fidelities together by the same amount) vs running only the epoch fidelity.

Figure 11 follows a similar structure as the heatmaps in Figure 5 but for ResNet-18 on CIFAR-100. We can draw similar conclusions as in Section 4.3 as we can achieve very high rank correlations with only a few trainable layers.

**Setup**. The setup mostly follows our previous experiments. For both fidelities we define the fidelity step size for layers (i.e., the discretization of the architectures) as described in Appendix F.1. For ResNet we set the step size for data as CIFAR-100 epochs with 20 being the full fidelity. For Transformers we set the maximum fidelity as 20 tokens per parameter following Hoffmann et al. (2022). To arrive at the data for Figure 5 and Figure 11, we log cost metrics and validation loss/error after each step or epoch for Transformers and ResNet, respectively.

| 9.15% Tokens | | | |
|---|---|---|---|
| Configuration | Rank Correlation | FLOPs [$10^{16}$] | Cost [in s] |
| Layer 1 (diagonal SH) | 0.26 | 0.22 | 2185.52 |
| Layer 10 (single SH) | 0.44 | 0.26 | 3467.48 |
| **19.21% Tokens** | | | |
| Configuration | Rank Correlation | FLOPs [$10^{16}$] | Cost [in s] |
| Layer 2 (diagonal SH) | 0.82 | 0.48 | 4843.13 |
| Layer 10 (single SH) | 0.84 | 0.55 | 7281.70 |
| **29.28% Tokens** | | | |
| Configuration | Rank Correlation | FLOPs [$10^{16}$] | Cost [in s] |
| Layer 3 (diagonal SH) | 0.82 | 0.75 | 8018.68 |
| Layer 10 (single SH) | 0.87 | 0.85 | 11095.92 |
| **39.43% Tokens** | | | |
| Configuration | Rank Correlation | FLOPs [$10^{16}$] | Cost [in s] |
| Layer 4 (diagonal SH) | 0.86 | 1.03 | 11392.41 |
| Layer 10 (single SH) | 0.87 | 1.14 | 14944.82 |
| **49.50% Tokens** | | | |
| Configuration | Rank Correlation | FLOPs [$10^{16}$] | Cost [in s] |
| Layer 5 (diagonal SH) | 0.87 | 1.32 | 15135.40 |
| Layer 10 (single SH) | 0.89 | 1.43 | 18759.04 |
| **59.56% Tokens** | | | |
| Configuration | Rank Correlation | FLOPs [$10^{16}$] | Cost [in s] |
| Layer 6 (diagonal SH) | 0.87 | 1.63 | 18892.12 |
| Layer 10 (single SH) | 0.88 | 1.72 | 22573.26 |
| **69.72% Tokens** | | | |
| Configuration | Rank Correlation | FLOPs [$10^{16}$] | Cost [in s] |
| Layer 7 (diagonal SH) | 0.87 | 1.94 | 23422.64 |
| Layer 10 (single SH) | 0.91 | 2.02 | 26422.16 |
| **79.78% Tokens** | | | |
| Configuration | Rank Correlation | FLOPs [$10^{16}$] | Cost [in s] |
| Layer 8 (diagonal SH) | 0.90 | 2.26 | 28088.84 |
| Layer 10 (single SH) | 0.93 | 2.31 | 30236.38 |
| **89.84% Tokens** | | | |
| Configuration | Rank Correlation | FLOPs [$10^{16}$] | Cost [in s] |
| Layer 9 (diagonal SH) | 0.92 | 2.60 | 32902.34 |
| Layer 10 (single SH) | 0.92 | 2.60 | 34050.61 |
| **100.00% Tokens** | | | |
| Configuration | Rank Correlation | FLOPs [$10^{16}$] | Cost [in s] |
| Layer 10 (diagonal SH) | 1.00 | 2.89 | 37899.50 |

Table 3: Quantifying markers from Figure 5 (14M GPT-2 model). The *single* SH represents a vanilla-SH run with epochs or update steps as the fidelity. For the *diagonal* SH run, we discretize the available layers in the same geometric pattern as recommended by SH. Thereby, we obtain a geometric progression of fidelity sources along both variables. At similar fidelity levels (layers-tokens), we see that both approaches provide similar rank correlations. However, training with fewer layers saves memory and that is seen in significant wall-clock time saving.

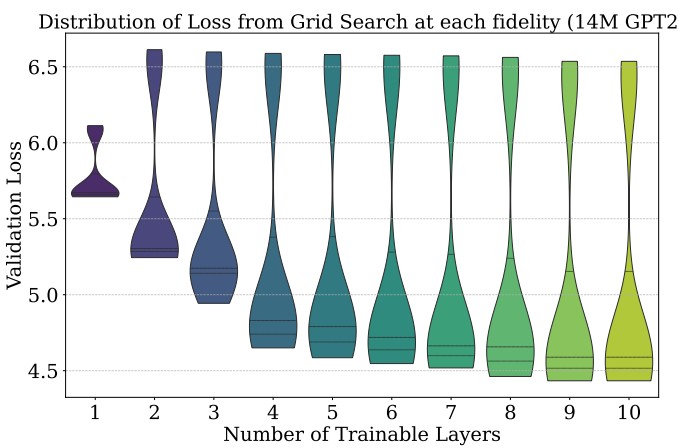

Figure 12: Loss distribution for the 14M GPT-2 across different number of trainable layers, given the search spaces from Table 1.

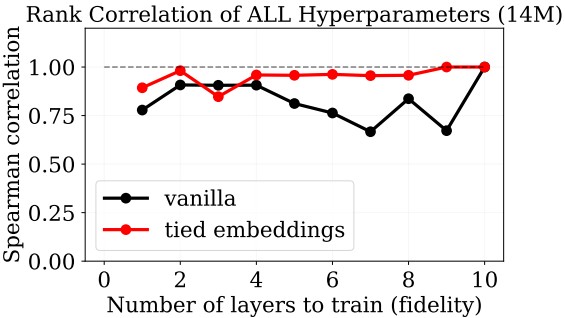

Figure 13: 14M parameter GPT-2 style Transformer comparison when embeddings are not shared (*vanilla*) or shared (*tied*). We follow previous results from the literature and conduct the majority of our experiments with tied embeddings (Zhong and Andreas, 2024). We detach the output tensor of the input embedding to retain the compute savings at low fidelities.

