# OpenReview forum: "Frozen Layers: Memory-efficient Many-fidelity Hyperparameter Optimization"
_automl.cc/AutoML/2025/Methods_Track — AutoML 2025 Methods Track_

### Official Review · Reviewer_qPMa · 2025-04-29

**Comments To Authors:**

The paper proposes to use the number of frozen layers during the training of DL models as fidelity in multi-fidelity HPO. Empirical evaluations across ResNets on a Computer Vision dataset and Transformer models on an NLP datasets show that the approach preserves the relative rankings of the low-fidelity models.

The paper is well written and the claims are supported by the experiments. The paper is highly relevant for the AutoML field in current times. It also has implications for the general large-scale training of LLMs, especially the finding that only unfreezing the final half of layers of a model can accurately predict hyperparameter candidate performance.

Strengths:
- The method is simple, yet the results are promising, essentially showing that the rankings of hyperparameters: type of optimiser, learning rate, weight decay, and the beta values of Adam are stable across the number of layers frozen
- Keeping more layers frozen results in lower compute resource usage in terms of GPU FLOPS and memory, meaning better efficiency and lower power consumption
- The approach can be combined with existing multi-fidelity HPO schedulers, such as Successive Halving

Weaknesses:
- 14M GPT2 models are rather small, e.g., the popular NanoGPT model [1] already has ~100M parameters
- There is a bump in Figure 4 on rank correlations for the GPT2 models at 30% layer freezing, which is not explained in the text
- It is not correct, that this paper proposes  “the first multi-fidelity source that opens the DL model training process itself, i.e., it modifies gradient computation and weight update steps”, as stated on page 2. Earlier works have already proposed to use the number of GPU for data-parallel training as a fidelity [1,2,3]. Increasing the number of GPUs also changes the training dynamics via a larger batch size and thus also influences gradient computation and weight updates [2,3,4].

As the approach is clear and the experimental results are promising I recommend the acceptance of the paper but would ask the authors to address the raised issues in the revision.

[1] https://github.com/karpathy/nanoGPT
[2] see Section 5.2 of https://arxiv.org/pdf/1810.05934
[3] https://dl.acm.org/doi/pdf/10.1145/3472883.3486989
[4] https://arxiv.org/pdf/2412.02729

**Review Confidence:**

5

**Review Rating:**

8

---

### Official Review · Reviewer_XsyX · 2025-04-30

**Comments To Authors:**

Summary: The authors propose a novel fidelity source for multi-fidelity hyperparameter optimization (MF-HPO) for the number of trainable layers in a model. By freezing a subset of layers during training, the authors demonstrate large savings in both compute and memory. This method has clear motivation and practical benefits. The claims are well supported by extensive empirical validation.

Strengths: The paper is well written and easy to follow. The idea of treating the number of trainable layers as a fidelity axis is highly novel as far as the reviewer is aware. The paper is clearly written and well-structured. Freezing layers to reduce memory is practical for efficiency. The authors conduct experiments across multiple architectures and fidelity levels, demonstrating clear trends in runtime, memory usage, and rank correlation. The method requires minimal architectural changes and can be easily integrated into existing training frameworks such as Megatron DeepSeep.


Weaknesses:

Lack of Ablation on Layer Selection Strategies:
 An empirical study on how different freezing strategies (e.g., freezing early vs. late layers) might impact the fidelity signal could be helpful. An ablation or sensitivity analysis can strengthen the paper.


Benchmark Scope is Limited:
 The empirical evaluation is comprehensive within the chosen architectures, but the reviewer would have liked to see results on at least one large-scale or real-world HPO benchmark, such as NAS-Bench, for comparison to other MF-HPO baselines.

The authors tackle a useful problem in a novel manner and practically useful. While some aspects could be improved, the empirical evidence is convincing. This paper can be of great interest to both AutoML and Deep Learning researchers.

**Review Confidence:**

5

**Review Rating:**

9

---

### Official Review · Reviewer_u7po · 2025-05-07

**Comments To Authors:**

- The authors present number of trainable layers as a new source of fidelity for hyper-parameter optimization for deep neural networks.
- The method is empirically evaluated on ResNet and Transformer (GPT2) family. The investigation on the considered examples has been thorough; however, the reviewer believes the number of architectures considered in this manuscript is not sufficient to draw any valid justifications. Validation on ResNet with Cifar100 and Transformer are two extreme cases. Addition of more architectures could have benefitted the work, particularly when the work depends heavily on the modular design of architecture. The authors could have considered different styles of networks to prove their claims.
-Specifically in ResNet and Transformer styled architectures, the task and model type exhibits consistent behavior across the depths. In these cases, the approach is bound to work successfully.
- It is not always a case where a shallow model and deep model captures the same patterns. In many situations, the shallow model underperforms than the deep model, leading to misleading fidelity estimates - how do authors address this concern?
- Layer count has been explored extensively as a fidelity dimension in NAS, particularly when the broad objective is to reduce the compute cost. How does it qualify as novel when similar metric is used in a more broader problem like HPO?
- It is intuitive that Cost monotonicity and Mutual Information Monotonicity will be satisfied for number of layers as fidelity for HPO - the reviewer do not see any significance in considering specifically these two as the criteria for judging the variable to be a valid fidelity parameter.
- The authors have not included any reasoning as to how and why only some specific layers should be selected or why only the first few layers should be frozen - why not last? selecting number of layers serves as valid fidelity, but why should they be only the last layers - This also raises question on whether the two criteria are sufficient to determine the validity of the variable as a fidelity in HPO.
-Training epochs and smaller subsets of data are also explored as variables for fidelity in HPO - the authors can compare with these or present an experiment where they are combining these with layer count for MF-HPO.
- There is no sound theoretical justification as to why layer count classifies as a measure of fidelity in HPO.
- The paper has been written well except for a few minor typos - authors may correct them.

**Review Confidence:**

5

**Review Rating:**

6

---

### Meta-Review · Area_Chair_6eCF · 2025-05-09

**Recommendation:** Accept
**Confidence:** 5

**Metareview:**

The paper proposes to use the number of frozen layers during training as fidelity in multi-fidelity HPO. This is a simple idea, but the reviewers are convinced by its potential practical impact. An ablation on the layer freezing strategy or the use of a wider set of network architectures could strengthen this work further.